# DDK: The Outsourced Kinase of Chromosome Maintenance

**DOI:** 10.3390/biology11060877

**Published:** 2022-06-07

**Authors:** Peter J. Gillespie, J. Julian Blow

**Affiliations:** Centre for Gene Regulation & Expression, School of Life Sciences, University of Dundee, Dundee DD1 5EH, UK; p.j.gillespie@dundee.ac.uk

**Keywords:** DDK, CDK, DNA replication, replication fork stability, DNA repair, chromatid cohesion, epigenetic inheritance

## Abstract

**Simple Summary:**

To ensure the maintenance of genetic stability prior to cell division a cell’s complement of chromosomes must be duplicated. This requires not only the replication of the chromosomal DNA but also the re-establishment the chromatin environment following duplication. To ensure the equal segregation of the genetic material to progeny cells, the duplicated chromatid pairs must remain physically coupled until cell division. The regulation of chromosome duplication is under the overall control of the cyclin-dependent kinases (CDK). In addition to maintaining global control of chromosome duplication, CDK directs the activation of a second kinase, the Dbf4-dependent kinase (DDK), which functions locally to facilitate the activation DNA replication and to coordinate this with the re-establishment of chromatin and the physical coupling of the chromatids following duplication. In this review, we discuss this ‘outsourcing’ by CDK to DDK of the activities that must be coordinated to ensure chromosome maintenance during cell division.

**Abstract:**

The maintenance of genomic stability during the mitotic cell-cycle not only demands that the DNA is duplicated and repaired with high fidelity, but that following DNA replication the chromatin composition is perpetuated and that the duplicated chromatids remain tethered until their anaphase segregation. The coordination of these processes during S phase is achieved by both cyclin-dependent kinase, CDK, and Dbf4-dependent kinase, DDK. CDK orchestrates the activation of DDK at the G1-to-S transition, acting as the ‘global’ regulator of S phase and cell-cycle progression, whilst ‘local’ control of the initiation of DNA replication and repair and their coordination with the re-formation of local chromatin environments and the establishment of chromatid cohesion are delegated to DDK. Here, we discuss the regulation and the multiple roles of DDK in ensuring chromosome maintenance. Regulation of replication initiation by DDK has long been known to involve phosphorylation of MCM2-7 subunits, but more recent results have indicated that Treslin:MTBP might also be important substrates. Molecular mechanisms by which DDK regulates replisome stability and replicated chromatid cohesion are less well understood, though important new insights have been reported recently. We discuss how the ‘outsourcing’ of activities required for chromosome maintenance to DDK allows CDK to maintain outright control of S phase progression and the cell-cycle phase transitions whilst permitting ongoing chromatin replication and cohesion establishment to be completed and achieved faithfully.

## 1. Introduction: Ensuring Chromosome Maintenance during the Mitotic Cell Division Cycle

In addition to the faithful duplication of the genome that occurs during S phase, the pre-replicative chromatin environment must also be re-established to maintain cell identity. Furthermore, in order to ensure the equal segregation of the duplicated genome to progeny cells, the replicated chromatid pairs must remain tethered prior to their anaphase segregation. The successful inheritance of a chromosome during cell division therefore requires the coordination of DNA replication and repair, chromatin re-formation and the establishment of chromatid cohesion. These activities should occur in concert to allow chromosome segregation and cell division during mitosis, all within a background of ongoing cellular processes, such as gene transcription, protein translation and energy metabolism.

The two kinases, cyclin-dependent kinase, CDK, and the Dbf4-dependent kinase, DDK, are required for DNA replication initiation (Figure 1 and Figure 2) [1,2,3,4]. CDK functions as the ‘global’ regulator of S phase entry and progression whilst ‘local’ control of replication origin activation is delegated to DDK [5]. This delegation of responsibility to DDK by CDK in higher eukaryotes is mediated by the CDK-dependent transcription of DDK and therefore its activation, at the G1-to-S transition [6,7,8]. In addition to its well characterised role in the initiation of DNA replication, it has been shown that DDK is required for chromatin re-formation following replication fork progression [9,10,11], for replication fork stability and DNA repair [12,13,14,15], and for the establishment of replicated chromatid cohesion [16,17] (Figure 1). The maintenance of chromosome stability during the mitotic cell-cycle therefore requires the coordination of CDK and DDK activity. This delegation of responsibility, or outsourcing, to DDK by CDK allows CDK to maintain outright control of S phase progression and the cell-cycle phase transitions whilst permitting ongoing chromatin replication and cohesion establishment to be completed and achieved faithfully.

## 2. Introducing the Dbf4-Dependent Kinase: DDK

DDK is formed by the association of the kinase subunit, cell division cycle 7 (Cdc7), with its activating partner, dumb-bell factor 4 (Dbf4) [18]. By analogy with CDK, Cdc7:Dbf4 is now most often referred to as ‘Dbf4-dependent kinase’ or DDK [19].

Whereas both proteins are well conserved across eukaryotes and orthologs have also been identified in plants, the nomenclature is not as well maintained between organisms (Table 1): with the exception of fission yeast in which the term hsk1 (homolog of seven kinase) is used, ‘Cdc7’ is remarkably well preserved; in contrast, orthologs of Dbf4 are variously known as dfp1 (dumb-bell factor *pombe*), him1 (hsk-interacting molecule) and rad35 in fission yeast, ASK (activator of S kinase) in human and chiffon in fruit fly. It should be noted, however, that the chiffon locus in fly is unique to that organism and encodes 2 polypeptides, chiffon A (Dbf4-like) and chiffon B (a Gcn5 acetylase binding protein) [20]. 

A second Dbf4 paralog, named either Drf1 (Dbf4-related factor 1) or Dbf4B, present in vertebrates including *Xenopus*, mouse and chicken, was first identified in human cells [21,22,23]. Immediately post-fertilisation, in developing *Xenopus laevis* eggs, DDK activity is predominantly supported by Drf1, although Dbf4 is also present but in substantially lower abundance [24]. On completion of the 12 reductive cleavage divisions immediately following fertilisation, eggs pass through the mid-blastula transition (MBT). At the MBT, gross reorganization of the cell-cycle occurs: bulk transcription begins, S phase and therefore cell-cycle length are extended and cell-cycle checkpoints are established [25,26,27]. Activation of Chk1 following the MBT promotes SCF^βTRCP^ degradation of Drf1, enabling the switch to Dbf4, which mediates the change in S phase kinetics [28].

In order to maximise comprehensibility in this review, we will, where possible, use the most commonly used names: Cdc7, Dbf4 and Drf1.

## 3. Cell-Cycle Regulation of DDK Activity

Early studies in budding yeast by Hartwell and colleagues of the execution point of the various cell division cycle mutants showed Cdc7 functioning downstream of the CDK kinase subunit Cdc28 and downstream of the ubiquitylation substrate adapter protein Cdc4; it was also reported that protein synthesis was required after these arrest points for the initiation of DNA replication to occur [29,30]. Akin to the cyclin:CDK analogy, it is the cell-cycle regulated abundance of Dbf4 that restricts the activation of DDK to the G1-to-S transition, with Cdc7 being present throughout the cell-cycle in this organism [31]. The periodicity of DDK activity is regulated not only by Dbf4 gene transcription, but also by cell-cycle-dependent protein degradation [32,33,34,35].

In contrast, in the human cell lines studied, it has been shown that the abundance of both Cdc7 and Dbf4/Drf1 protein levels are periodic [36,37]. The peak of Dbf4 (and also Drf1) mRNA is in the late G1 and S phase [21,38,39]. In mammalian cells, both Cdc7 and Dbf4/Drf1 are transcriptional targets of the E2F family of transcription factors, activated following satisfaction of the restriction point by either pRb inactivation, degradation or both, upon activation of CDK during G1 [6,7,8]. Consistent with this, Cdc7 and Dbf4/Drf1 proteins levels rapidly increase as cells enter S phase and high levels are maintained in G2 and early M [21,36,37]. That CDK is required for the transcriptional activation of its partner S phase kinase DDK in human cells illustrates the hierarchical nature of their relationship (Figure 1). 

Following passage of the metaphase-to-anaphase transition and through until mid-G1, Cdc7 protein levels in human cells and Dbf4/Drf1 protein levels in both yeast and human return to their initial low levels [32,34,35,36,37]. This stark drop in levels of the kinase activating subunit in budding yeast is mediated by the APC/Cyclosome (APC/C) [33,34,35]. Consistent with this, levels of Dbf4 remain low in α-factor (G1) arrested budding yeast and in quiescent human cells [21,33,34,35,39]. Inactivation of the APC/C in G1 mediated by G1/S CDK and Skp1:Cullin1:F Box:Cdc4 allows the accumulation of DDK activity as budding yeast cells pass into S phase. In human cells, inactivation of APC/C:Cdh1 by Chk1 following replication stress stabilises both Cdc7 and Dbf4 suggesting that DDK is also a target of Cdh1 in higher eukaryotes [40].

## 4. Replication Origin Priming: DDK-Mediated Phosphorylation of MCM2-7 

During late mitosis (telophase) and G1, potential replication origins are first loaded with inactive double hexamers of the MCM2-7 proteins to license them for a single initiation event in the upcoming S phase [1,2,4]. DDK plays a key role in phosphorylating MCM2-7 double hexamers to drive their transformation into the replicative CMG (Cdc45:MCM:GINS) helicase. Initial studies of Cdc7 and Dbf4 function in yeast identified MCM subunits as suppressors of DDK temperature sensitive mutations and localized DDK to replication origins [41,42]. Subsequent studies in human cells provided the first direct link between DDK and MCM2-7 phosphorylation by showing in vitro that 2 MCM subunits were DDK substrates [43]. In budding yeast, human and in *Xenopus* cell-free extracts, phosphorylation of MCMs 2, 4 and 6 have been shown to be DDK-dependent [12,44,45,46,47,48]. It remained unclear, however, which of these were the key physiological targets of DDK to support DNA replication in vivo. 

In budding yeast, in vivo, the N-terminus of MCM4 is hyperphosphorylated by DDK [45]. An autoinhibitory activity present in the N-terminus of MCM4 is relieved upon DDK phosphorylation and in yeast cells lacking this region, DDK is not required for cell viability [45]. This suggests that relief of MCM4 autoinhibition by its phosphorylation is the essential function of DDK in this organism to support DNA replication initiation. The precise role of the phosphorylation of the other MCM subunits remains to be determined but may be important for other functions of DDK in maintaining chromosome stability (see below).

Consistent with the identification of MCM4 as the key target of DDK-mediated phosphorylation in budding yeast, it was found that in both human cells and in the *Xenopus* cell-free system that hyperphosphorylation of MCM4, but not the phosphorylation of specific MCM2 residues, correlates with replication initiation [12,49]. In these studies, MCM4 hyperphosphorylation occurred only on chromatin-bound double hexameric MCM2-7 at licensed replication origins. In contrast, phosphorylation of two known DDK target residues on MCM2 (S40 and S53) occurred on single hexamers free in the nucleoplasm and could be inhibited at concentrations of DDK inhibitors that do not inhibit DNA replication. This suggests that the MCM2-7 double hexamer at licensed replication origins provides the key substrate for DDK to support replication initiation.

The recent description of structural models of DDK associated with chromatin-bound MCM2-7 has highlighted the key role of double hexamer formation on chromatin in promoting the activating phosphorylation of MCM4 [50,51,52]. Dbf4 docks onto one MCM2-7 hexamer to support the phosphorylation of the other hexamer (Figure 2ii, detail). Loss of the Dbf4 docking domain abrogates phosphorylation. The rotationally symmetrical nature of the hexamer explains the ability to phosphorylate both MCM4 subunits at the origin. The docking of Dbf4 to the double hexamer is mediated by an N-terminal BRCT (BRCA1 C-terminal) phosphobinding motif; this motif is conserved in higher eukaryotic Dbf4 orthologs. The docking site for the Dbf4 BRCT domain on the hexamer is the phosphorylated N-terminus of MCM2, although the phosphorylating kinase remains to be determined. Two candidate kinases are CDK and DDK itself: it may be that the previously described DDK-mediated phosphorylation of S40 and S53 of MCM2 in single MCM2-7 hexamers promotes this association on formation of the double hexamer on chromatin; alternatively, or in addition to this, CDK may mediate this phosphorylation to signal fitness for S phase entry. 

The phosphorylation of MCM4 by DDK is required to support the localization of key replication proteins to MCM2-7 [1,2,4] (Figure 2). In budding yeast, DDK phosphorylation of MCM2-7 promotes the recruitment of the Sld3:Sld7 complex and Cdc45 to origins. The combination of DDK activity and CDK, which phosphorylates Sld3 and Sld2, promotes the association of Dpb11 and Sld2 with Sld3:Sld7, facilitating the association of the GINS complex and DNA polymerase ε to origin sites. In combination, these factors recruited to MCM2-7 at licensed origins, together with MCM10, promote the formation and separation of the 2 CMG replicative helicases that then function to support DNA unwinding, replication initiation and elongation. Since loss of the auto-inhibitory N-terminal domain of MCM4 in budding yeast permits replication initiation in the absence of DDK and mutation of MCM5 supresses the requirement for DDK in replication initiation, this suggests that MCM2-7 and possibly just MCM4, are the only essential DDK substrates for the initiation of replication in budding yeast.

The role of DDK-mediated phosphorylation of the chromatin-bound MCM2-7 double hexamer in promoting replication initiation has been shown to be substantially conserved in higher eukaryotes (Figure 2). CDK-dependent phosphorylation of Treslin (TopBP1-interacting, replication-stimulating protein), the metazoan ortholog of Sld3, is required for its association with TopBP1, the metazoan ortholog of Dpb11 [53,54,55]. In the *Xenopus* cell-free system, the association of chromatin-bound MCM2-7 double hexamers with the Treslin:MTBP (MDM2-binding protein) complex, the ortholog of the budding yeast Sld3:Sld7 complex, is both increased and strengthened following DDK phosphorylation of MCM4 [5].

## 5. Rif1: Antagonising DDK-Mediated MCM4 Phosphorylation

Whereas DDK mediates MCM4 phosphorylation to support replication initiation, recent studies have identified a role for protein phosphatase 1 (PP1) in the reversal of this [47,56,57,58] (Figure 2i). In human cells and in *Xenopus* egg extract, PP1 continually dephosphorylates MCM4 such that inhibition of DDK mid-S phase inhibits further replication initiation [12,47,59]. In yeast, human cells and in the *Xenopus* cell-free system, PP1 is targeted to MCM2-7 on chromatin by Rif1 (Rap1-interacting factor 1) [12,56,59]. Rif1 was first identified as an antagonist of Rap1-mediated gene silencing [60]. PP1 binds Rif1 by the PP1-targeting motif RVxF [61,62]. Mutation of these residues or loss of Rif1 leads to the increased activation of replication origins and compensates for a reduction in DDK activity, facilitating the shortening of S phase, showing that whereas DDK promotes S phase progression, Rif1 counteracts this [12,13,56,59].

Loss of Rif1 or the Rif1:PP1 interaction also perturbs the replication timing programme [12,63,64]. This result is consistent with the idea that selection of origins for DDK-dependent phosphorylation forms part of the mechanism that enacts the replication timing programme. In addition to replication timing, Rif1 has been shown to impact nuclear architecture, limiting interaction between domains with the same replication timing [64]. Both nuclear organization and replication timing depend upon Rif1:PP1 interaction; however, timing can be established and executed independent of a specific 3D genome organization or spatial distribution of replication foci [65]. This suggests that coregulation of replication timing and genome organization is specifically linked to Rif1 and not generic nuclear architecture.

In addition to this, Rif1:PP1 has been found to function in a number of DNA replication-associated processes including telomere length regulation, DNA repair and the promotion of replication licensing in G1 by stabilizing Orc1 [59,60,66]. 

It should be noted that a recent study describes the growth of both human and murine cells in which DDK activity is either inhibited or depleted [67]. Murine cells harbouring an ‘analogue sensitive’ mutation in the Cdc7 ATP binding pocket showed retarded proliferation but did not arrest following treatment with an ATP analogue; furthermore, following either chemically- or genetically-mediated degradation of either Cdc7 or Dbf4 (a requirement for Drf1 was not tested), both human and murine cells can maintain proliferation although changes to S phase length were apparent. The authors conclude that DDK activity is not strictly necessary for proliferation and that it is functionally redundant with CDK. However, although DDK proteins levels were significantly reduced following induced degradation the targeted proteins remained detectable and furthermore, although the phosphorylation of a specific MCM2 residues was lost upon degradation, the phosphorylation of residues in the N-terminal region of MCM4, which better correlates with replication initiation, was still apparent. Thus, it may be the case that proliferation in these cells is the result of a combination of the remaining DDK activity and the maintained phosphorylation of DDK target sites, perhaps following the inactivation of Rif1:PP1 by CDK [68]. Consistent with this, the loss of Rif1 reduces the sensitivity of cells to DDK inhibition [69]. In order to clarify these issues, further work is needed to determine the essential phosphorylation sites required for cell proliferation and which kinases can phosphorylate them.

## 6. Replication Origin Selection and Activation: DDK-Mediated Phosphorylation of Treslin:MTBP

An additional role for DDK activity in the initiation of replication has also been demonstrated in the *Xenopus* system [5] (Figure 2)**.** Independently of MCM phosphorylation, DDK activity has been shown to promote the formation of Treslin:MTBP:TopBP1 complex that supports replication initiation (Figure 2iv,v). The DDK substrate mediating this increased complex formation remains to be fully characterised as both Treslin and MTBP demonstrate a DDK-dependent mobility shift [5,70]. Furthermore, it was shown that DDK-dependent phosphorylation of Treslin is required to facilitate the conserved CDK-mediated phosphorylation of Treslin that is required for interaction with TopBP1 [5]. This suggests that in addition to priming replication origins by mediating phosphorylation of MCM2-7, DDK facilitates formation of the Treslin:MTBP:TopBP1 phospho-MCM2-7 reader complex in higher eukaryotes. In budding yeast, Sld3 has also been shown to be subject to DDK dependent phosphorylation but the functional consequence of this remains to be determined [58].

Consistent with the role of Rif:PP1 in reversing the phosphorylation of MCM2-7 double hexamers at origins, in budding yeast, Rif1 was found to reverse phosphorylation of Sld3 [58]. Although the role of Rif1 in the *Xenopus* cell-free system has not been confirmed, PP1 was found to reverse the DDK mediated phosphorylation of Treslin and moreover to inhibit formation of the Treslin:MTBP:TopBP1 complex [5]. It remains to be determined if the CDK-mediated phosphorylation of Treslin/Sld3 is also subject to dephosphorylation, e.g., by PP2A, but this would be expected (Figure 2vi).

## 7. Which Acts First: CDK or DDK?

This fuller understanding of the functions of CDK and DDK during the initiation of replication can start to resolve the long-standing question of which of the two kinases, CDK or DDK, functions first in promoting S phase entry. CDK activity must precede DDK in organisms in which CDKs are required to promote the transcription of Cdc7 and Dbf4. DDK activity is then required for phosphorylation of origin-bound MCM2-7 double hexamers, but the recruitment of DDK to double hexameric MCM2-7 requires phosphorylation by unknown kinases that could also include CDK and DDK. In addition, CDK (and in *Xenopus*, DDK) activity is required to promote the formation of the Sld3:Sld7:Dpb11 complex (yeast) or Treslin:MTBP:TopBP1 complex (vertebrates). Rather than one kinase functioning exclusively before the other, it is therefore the integration of CDK and DDK activity on multiple substrates that promotes replication initiation. This coordination demonstrates the distinct ‘global’ and delegated ‘local’ roles that CDK and DDK, respectively, play in DNA replication initiation. By regulating the abundance of DDK, CDK initiates a self-governing feedback loop for replication initiation in which it executes the ultimate activation of replication origins primed by DDK (Figure 1). 

## 8. Replication Origin Selection

In human cells, the number of origins licensed in G1 is in two-to-threefold excess over the number of origins that are activated in an unperturbed S phase [71]. There is a great deal of stochasticity in which of the licensed origins are actually selected for initiation in any given cell passing through S phase. The otherwise dormant excess origins can be activated when replication forks stall and thus provide a means to maintain replication rates under conditions of replicative stress [71]. The number of potential replication origins is also greater than the quantity of the limiting origin activating proteins: Treslin (Sld3 in yeast), Dbf4/Drf1, TopBP1 (Dpb11 in yeast) and RecQ4 (probable ortholog of yeast Sld2). In human cells, only a fraction of chromatin-bound MCM4 molecules are phosphorylated [12], consistent with the idea that DDK-dependent phosphorylation of MCM2-7 double hexamers at licensed origins could play a role in selecting which particular origins fire during S phase. 

The DDK-dependent loading of Treslin:MTBP onto MCM2-7 is the earliest known event that selects which origins will initiate. To maintain genome stability, it is necessary to regulate the distribution of active replication forks in response to replication stress [72]. The correct selection of potential origins at which to initiate replication is therefore of critical importance. With PP1 able to reverse both the activating phosphorylation on MCM2-7 and Treslin:MTBP, the local balance of DDK and CDK activity with that of the respective phosphatase activity must therefore be coordinated to support origin activation. 

The local chromatin environment is likely to play an important role in origin selection. In human cells, BRD2 and BRD4, G-quadruplex DNA, the AP-1 binding motif and open chromatin linked histone post-translational modifications have been shown to promote association of Treslin to specific genome sites [73,74]; in budding yeast, the forkhead transcription factors localise DDK to early origins [75]. Furthermore, the transcription-coupled kinase, Cdk8/19:cyclin C has been shown to interact with MTBP in human cells and in this way, the integration of other chromatin associated activities, such as transcription, with replication initiation proteins may determine preferred sites of origin activation [76]. Since local active replication origin density is limited by the ATR-Chk1 pathway [77,78], which functions by countering the CDK-dependent destabilization of Rif1:PP1 [68], the origin selection process not only determines which origins will activate but also those that cannot, further emphasising the significance of origin selection.

## 9. Beyond Replication Initiation: The Role of DDK in Replication Fork Stability during Replication Stress 

During replication elongation, the progress of the replication fork can be stalled upon encountering a site of DNA damage or a protein barrier. To ensure the resumption of replication elongation following resolution of the lesion it is necessary to maintain the integrity of the fork as the loss of key replisome proteins or disengagement from the DNA may facilitate fork instability, ultimately leading to fork collapse and the inability to restart the stalled fork. The phosphorylation of MCM4 has been shown to be maintained in the CMG helicase during replication elongation [12,13,46]. The maintenance of MCM4 phosphorylation in the CMG correlates with replication fork stability and both DDK and Chk1 are required to prevent fork collapse [12,13,69]. As with MCM2-7 double hexamers and Treslin:MTBP prior to origin activation, the phosphorylation of MCM4 in CMG is subject to dephosphorylation by PP1. The kinetics of the dephosphorylation of MCM2 and MCM4 by PP1 is different between unactivated double hexamers and CMG: whereas at licensed origins both MCM2 and MCM4 are dephosphorylated by Rif1:PP1, MCM4 but not MCM2 in the CMG remain partially phosphorylated (Figure 3) [12]. In the absence of Chk1, Rif1:PP1 drives replisome instability in the presence of a range of agents that promote replication stress, including aphidicolin, etoposide and UV [12]. This Rif1:PP1-induced replisome instability is enhanced by DDK inhibition. These results are consistent with the idea that, similar to the situation at licensed origins, DDK-mediated MCM phosphorylation promotes CMG stability and that this is reversed by Rif1:PP1. The replisome stability function of Chk1 therefore becomes critical because of the action of Rif1:PP1 in reversing DDK-dependent phosphorylation [12]. Furthermore, loss of either Rif1 or ETAA1 (an activator of the ATR-Chk1 pathway monitoring the completion of S phase) reduces the sensitivity to inhibition of DDK, confirming that both DDK and Chk1 oppose the replisome-destabilising action of Rif1:PP1 [69]. Consistent with the role of DDK in maintaining fork progression, inhibition of DDK in human cells in the absence of exogenous stress causes stalling of elongating forks, and the stabilisation and restart of these forks is DDK-dependent [13].

It is unknown whether the DDK phosphorylations that maintain replisome stability are simply the ones put on MCM2-7 double hexamers or whether new phosphorylation sites are involved. Either way, the continuing presence of DDK in the replisome may be critical to ensuring continued replication elongation in both the absence and presence of replication stress. As described above, DDK-mediated phosphorylation of MCM4 prior to replication initiation requires the formation of the MCM2-7 double hexamer to support Dbf4 docking across the hexamers and this docking site is lost upon origin activation and splitting of the hexamers on formation of the CMG. This suggests that association of DDK with the elongating replisome would be dependent on another targeting component. 

It has been shown that the budding yeast ortholog of the checkpoint adaptor protein Claspin, together with the Timeless:Tipin orthologs, is required to maintain coupling between the CMG helicase and the replisome [79,80,81]. Both budding yeast and human Claspin are required to regulate replication fork speed [82,83,84]. The initial association of Claspin with chromatin in the *Xenopus* cell-free system and in budding yeast is DDK dependent, consistent with Claspin’s chromatin association being dependent on replication initiation [23,85,86]. In human cells, Claspin localises DDK to the replisome and mediates phosphorylation of both Claspin itself and the MCM proteins [87,88,89]. DDK-mediated phosphorylation reduces an intramolecular interaction within Claspin which unmasks both a DNA binding and a PCNA binding PIP box, facilitating their interaction and promoting interaction with MCM2 [89]. Although structurally very different to the chromatin-bound double hexameric MCM2-7 complex, it may again be MCM2, in this case via Claspin, that directs DDK towards MCM4 to promote its continued phosphorylation. Claspin’s role as a checkpoint adapter protein, in which inactive Chk1 localises to Claspin to facilitate Chk1’s activation by ATR, is dependent upon the phosphorylation of Claspin’s Chk1 binding domain (CKBD); it has been shown that whereas casein kinase 1 can perform this phosphorylation in normal cells, DDK can perform this Claspin phosphorylation in cancer cells, thereby facilitating Chk1 activation [90,91] In combination, these results highlight the key role of DDK in mediating the response to replication stress.

Following replication stress, e.g., upon DNA damage, as well as maintaining replication fork stability, the primary lesion must be attended to. In human cells this requires the activation of the checkpoint signalling proteins ATR or ATM, which both prevent fork collapse and initiate processing of the problematic lesion. In budding yeast, temperature sensitive alleles of both DDK subunits significantly reduce the rate of mutagenesis in response to DNA damaging agents [92,93]; conversely, in human cancers, DDK overexpression is correlated with higher mutation frequencies [8]. These studies suggest that DDK plays a role in the actual processing of the DNA lesion. 

There are several known examples of DDK-dependent phosphorylation activating DNA processing enzymes. In human cells, it has been shown that DDK is required to promote the processing of stalled replication forks to initiate checkpoint signalling in response to hydroxyurea (HU) [14]: in the absence of DDK activity checkpoint activation and fork processing are defective, leading to genome instability. This study suggests that the likely phosphorylation target of DDK is nuclease EXO1, which is required for nascent strand degradation. In budding yeast, the activation of the Mus81:Mms4 resolvase during mitosis, which promotes the disentanglement of joint DNA molecules, is dependent on its phosphorylation by DDK [94]. In human cells, the phosphorylation of RAD18 by DDK in response to UV light promotes its interaction with DNA polymerase η to facilitate trans-lesion DNA synthesis [95].

DDK also plays a role in promoting fork restart after forks have stalled or reversed. Following a fork stall caused by topoisomerase inhibition, DDK activity is required for the excessive MRE11-mediated DNA degradation that occurs in the absence of BRCA2, suggesting that DDK activates and BRCA2 limits the activity of MRE11 [15] In this study the phosphorylation of MRE11 was reduced upon DDK inhibition, suggesting that MRE11 is a DDK substrate and therefore that DDK directly activates MRE11. Significantly, DDK was found to localise to stalled forks and was shown to be required for both fork slowing and stabilisation and importantly for fork restart. Considering the localization of DDK to these stalled forks, it is tempting to speculate that this may be mediated by the interaction of DDK with Claspin as described above.

Rif1:PP1, which reverses DDK-dependent MCM phosphorylation at licensed origins, is also enriched at stalled replication forks. In both budding yeast and human cells, excessive degradation of reversed forks is seen in the absence of Rif1 [96,97,98,99]. However, treatment of Rif1-depleted human cells with an inhibitor of DDK did not reverse the DNA resection following HU treatment [97]. This does not necessarily rule out DDK as the kinase activity reversed by Rif1:PP1, as the phosphorylation of existing DDK targets involved in the processing of stalled forks may persist. 

Taken together, these results highlight the complex interplay between DDK and Rif1:PP1 in the multistep processes that ensure the induction of checkpoint signalling and the processing, stabilisation and restart of stalled replication forks following DNA damage. Whereas DDK is required to maintain the viability and ensure restart of stalled forks, Rif1:PP1 may play a role in restricting fork progression following checkpoint activation by dephosphorylating MCM subunits in the CMG. The downstream processing of forks in response to replication stress also requires active DDK. Although activity of DDK in both budding yeast and human cells has been shown to be restricted upon phosphorylation of Dbf4 by the checkpoint kinases, this inhibition may not be absolute and more importantly may be localised [13,34,100,101]. Further to mediating the inhibition of DDK, checkpoint kinase mediated phosphorylation of, or association with, Dbf4 could also direct the kinase away from substrates required for replication initiation to those associated with fork processing, thus ensuring fork integrity. Consistent with this, in budding yeast, in addition to phosphorylating Dbf4 to inhibit DDK binding to licensed origins, rad53 is able to block DDK’s association with MCM2-7 independent of its kinase activity [50,102]. In the *Xenopus* cell-free system the balance of DDK activity is reduced following replication stress [47,103], possibly through checkpoint-mediated direct inhibition of DDK or by the stabilisation or relocalisation of the Rif1:PP1 complex [68]. Further investigation is required to fully characterise the local changes in DDK kinase and Rif1 phosphatase activity that mediate fork stability, repair and restart.

## 10. DDK and Post-Replicative Chromatin Formation

As the genome undergoes duplication during S phase, it is not only the DNA that must be replicated, but also the underlying chromatin microenvironment must be re-established, including nucleosome formation and the pattern of histone modifications, as well as transcription factor chromatin association. The maintenance of this epigenetic environment during genome duplication is therefore required to maintain the pre-replicative cellular state and ultimately cellular identity [104,105]. However, in circumstances in which a directed change in cellular identity is required, for example, when the pluripotent state of cells is lost upon differentiation, the chromatin microenvironment must be altered. S phase may thus provide an opportunity to facilitate a change in cellular identity during differentiation by altering the incorporation of specific histone variants or mediating changes in their post-translational modification and also by regulating the availability of differentiation regulated transcription factors. It has been shown that change in the speed of replication fork passage is critical during differentiation consistent with the key role of the replication process in facilitating changes in cellular identity [106,107].

Following passage of the replication fork, the re-formed chromatin is composed of both newly synthesized histones and the transferred parental histone subunits. Whilst the transfer of the parental histones to the newly synthesized DNA can re-establish the pre-replicative chromatin environment, the deposition of newly synthesized histones might alter this.

DDK has been shown to have a role in post-replicative chromatin formation (Figure 4). In both budding yeast and in higher eukaryotes, DDK has been shown to phosphorylate CAF1 (chromatin assembly factor 1) (Figure 4i) [9,10,11]; CAF1 binds PCNA and deposits H3:H4 tetramers behind the replication fork on newly synthesized DNA [108,109,110]. In budding yeast, mutation of the DDK phosphorylation sites of the CAC1 subunit of CAF1 leads to loss of gene silencing consistent with previously described defects in gene silencing in DDK mutants [10,11].

Histone binding motifs have been identified in multiple replisome components including the MCM2 subunit of the CMG helicase and two DNA polymerases, α and ε [111,112,113,114,115]. In both budding yeast and human cells it has been shown that MCM2 functions as a histone chaperone distributing parental histone H3:H4 tetramers to the newly synthesized DNA on the lagging strand (Figure 4ii) [116,117]; distribution of the tetramers to the leading strand is facilitated by DNA polymerase ε (Figure 4ii) [118]. It is interesting to note that histones H3:H4 are chaperoned by MCM2. MCM2 has previously been identified as a target of DDK-mediated phosphorylation; it therefore remains an interesting possibility that DDK may regulate MCM2 chaperone activity.

## 11. DDK and The Establishment of Replicated Chromatid Cohesion

The establishment of replicated chromatid cohesion during S is absolutely required to maintain genome stability: in the absence of cohesion chromatids separate prematurely, prior to the metaphase-to-anaphase transition, resulting in defects in metaphase plate formation and ultimately chromosome mis-segregation [119,120].

The tethering of replicated chromatids during S phase is mediated by cohesin [119,121]. Cohesin is loaded onto chromatin by the cohesin loading complex Scc2:Scc4, which clamps the cohesin ring around DNA [122,123,124,125]. In the *Xenopus* cell-free system and in human cells, the establishment of cohesion during S phase has been shown to be coupled to replication origins by the MCM2-7 proteins and DDK [16,17,123,124]. In both systems the association of Scc2:Scc4 with DNA and cohesin loading requires the licensing of replication origins by MCM2-7 and furthermore the activity of DDK. Scc2:Scc4 and cohesin both associate with DDK in both systems and in the *Xenopus* cell-free system the association of DDK and cohesin was found to be dependent on Scc2:Scc4 [16,17]. Despite investigations in both systems the target of DDK-mediated phosphorylation still remains to be determined as it is difficult to parse the known effect of DDK loss or inhibition on MCM2-7 phosphorylation with the potential role of DDK in phosphorylating additional proteins, such Scc2:Scc4 or cohesin itself; this is especially difficult in human cells as the late telophase and G1 association of cohesin with chromatin, when cohesin functions to shape the genome and in transcriptional regulation, is independent of MCM2-7 [126]. 

Since DDK is required both for the activation of replication origins and for mediating cohesin chromatin association, it is ideally positioned to play a key role in ensuring the coordination of the two processes.

## 12. Therapeutic Potential of Targeting DDK

The overexpression of DDK in cancer cells, found in ~50% of lines tested, correlates with both increased chemoresistance and mutation frequencies [8,127]. High-level Cdc7 and Dbf4 overexpression is associated with inactivation of the Rb pathway, consistent with both DDK genes being transcriptional targets of E2F and also with inactivation of p53 [8,127]. Considering the multiple roles that DDK plays in maintaining genome stability, it is a very attractive target in the development of anti-proliferative chemotherapeutics. At this time, three such agents have been developed and have undergone clinical trials but as yet no compound has reached the market [128,129,130]. All three compounds, PHA-767491, XL413 and TAK-931/simurosertib are competitive inhibitors of ATP binding to DDK. Two further compounds, LY3143921 and TQB3824, are currently undergoing clinical trials. Given that DDK’s role in the initiation of DNA replication precedes its other known roles, it might be expected that any antiproliferative effects might be mediated by inhibiting the initiation of DNA replication. However, since small molecule inhibition of DDK does not prevent its association with its target proteins, e.g., inhibited DDK still binds to MCM2-7 on chromatin, it may be that the maintained association of inhibited DDK with alternative very low abundance fork stability and DNA repair substrates that mediate most clinical efficacy. In this case, using an anti-DDK therapeutic as a low dose secondary agent in combination with a primary chemotherapeutic may be the most effective strategy to potentiate efficacy.

Considering the known and potential functions of DDK in multiple aspects of maintaining genetic stability and importantly the coordination between these (Figure 1), further investigation of the balance of local DDK to Rif1 activity and the effects of DDK inhibition in both normal and cancerous human cells is required to reveal the full potential of targeting it in the clinic.

## 13. Conclusions

In addition to its now well characterised role in phosphorylating the MCM2-7 double hexamer at replication origins to facilitate the initiation of DNA replication, the role of DDK in the regulation of several additional functions required for chromosome duplication has been reported. These additional functions include the maintenance of replication fork stability following replication stress, the re-establishment of chromatin and the establishment of chromatid cohesion following DNA replication. Furthermore, a second DDK-dependent step has been described in the initiation of DNA replication, the formation of Treslin:MTBP:TopBP1 complex, thus identifying an additional point at which replication origin activation can be regulated.

The regulation of DDK activation by CDK in higher eukaryotes provides a means by which CDK can act as the ‘global’ regulator of S phase whilst ‘local’ control is delegated to DDK. This ‘outsourcing’ of activities required for chromosome maintenance to DDK allows CDK to maintain outright control of S phase progression and the cell-cycle phase transitions whilst permitting ongoing chromatin replication and cohesion establishment to be completed and achieved faithfully.

## Figures and Tables

**Figure 1 biology-11-00877-f001:**
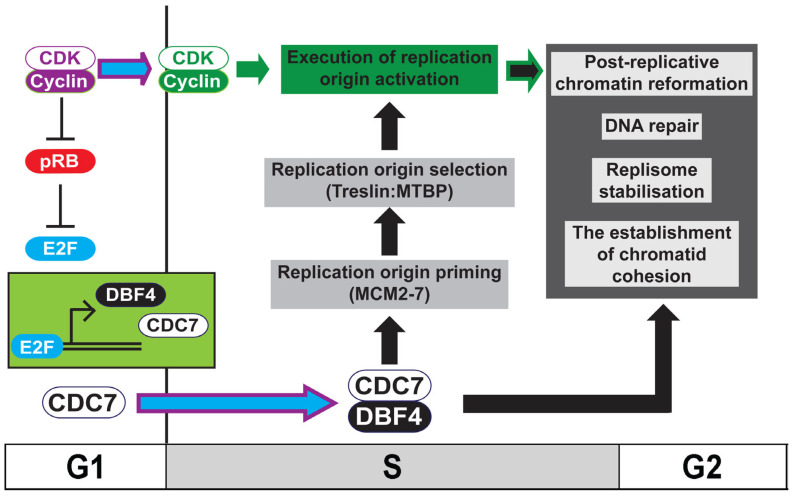
Model for the role of DDK in ensuring chromosome stability. Activation of Cdc7 and Dbf4 transcription by E2F in higher eukaryotes occurs following the inactivation of pRB by rising CDK levels at the G1-to-S transition. During S phase, the activation of replication origins requires the coordination between active DDK (Cdc7:Dbf4), which both primes (MCM2-7 phosphorylation) and directs origin selection (Treslin:MTBP phosphorylation) and CDK, which executes replication initiation. In addition to its well-characterised role in replication initiation, DDK is required for the establishment of replicated chromatid cohesion, the maintenance of replisome stability and DNA repair, and is required for the re-formation of chromatin following replication of the DNA. The coordination between CDK and DDK therefore ensures chromosome maintenance during the mitotic cell-cycle.

**Figure 2 biology-11-00877-f002:**
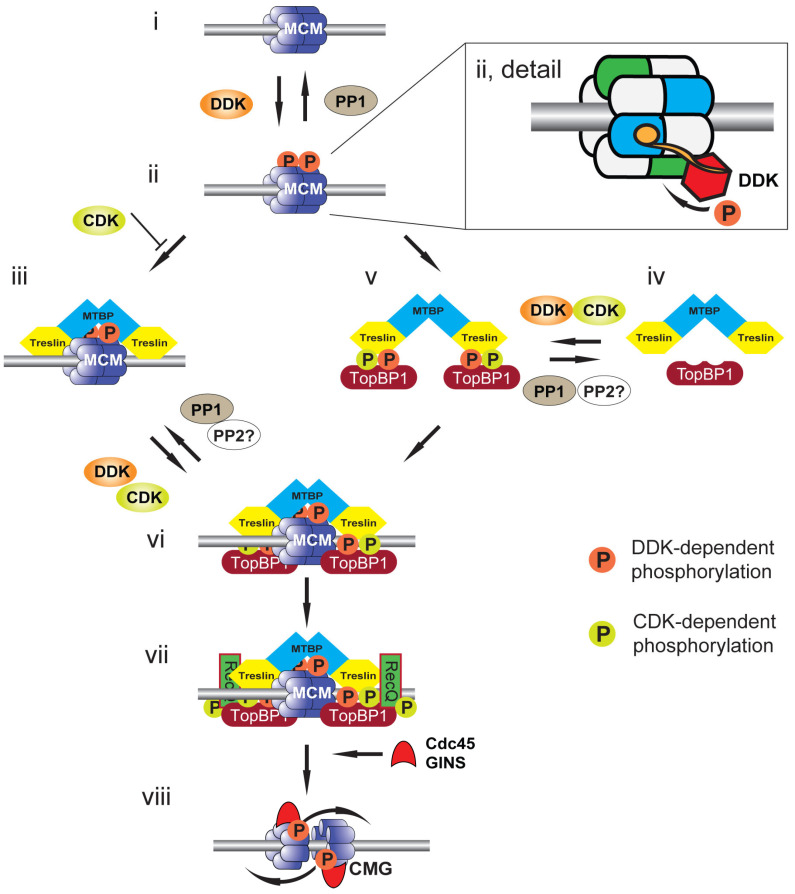
Model for CMG formation and activation. Cartoon illustrating events at a replication origin leading to replication initiation. (**i**) Replication origin DNA (grey bar) licensed by DNA-bound MCM2-7 double hexamer (blue). (**ii**) DNA-bound MCM2-7 is phosphorylated by DDK (orange circles) and phosphorylation is reversed by PP1 recruited to origins by Rif1. Detail shows a DNA-bound MCM2-7 double hexamer in which the BRCT phosphobinding-domain of Dbf4 (orange circle) is bound to MCM2 (blue) of one hexamer to localise Cdc7 (red) to phosphorylate MCM4 (green) of the second hexamer. (**iii**) Treslin:MTBP is recruited to DDK-phosphorylated MCM2-7. This interaction is opposed by CDK (green circles), most likely due to CDK phosphorylation of Treslin. (**iv**) Soluble Treslin:MTBP can be phosphorylated by both CDK and DDK. (**v**) Phosphorylated Treslin:MTBP binds to TopBP1 in solution. (**vi**) The Treslin:MTBP:TopBP1 complex is recruited to DDK-phosphorylated MCM2-7. (**vii**) RecQ4, the second CDK substrate required for replication initiation, is recruited to MCM2-7:Treslin:MTBP:TopBP1 to form the pre-initiation complex (pre-IC). (**viii**) The CMG helicase is formed upon Cdc45 and GINS recruitment to MCM2-7. The MCM2-7 double hexamer is split and the helicases overtake one another to support bidirectional DNA replication.

**Figure 3 biology-11-00877-f003:**
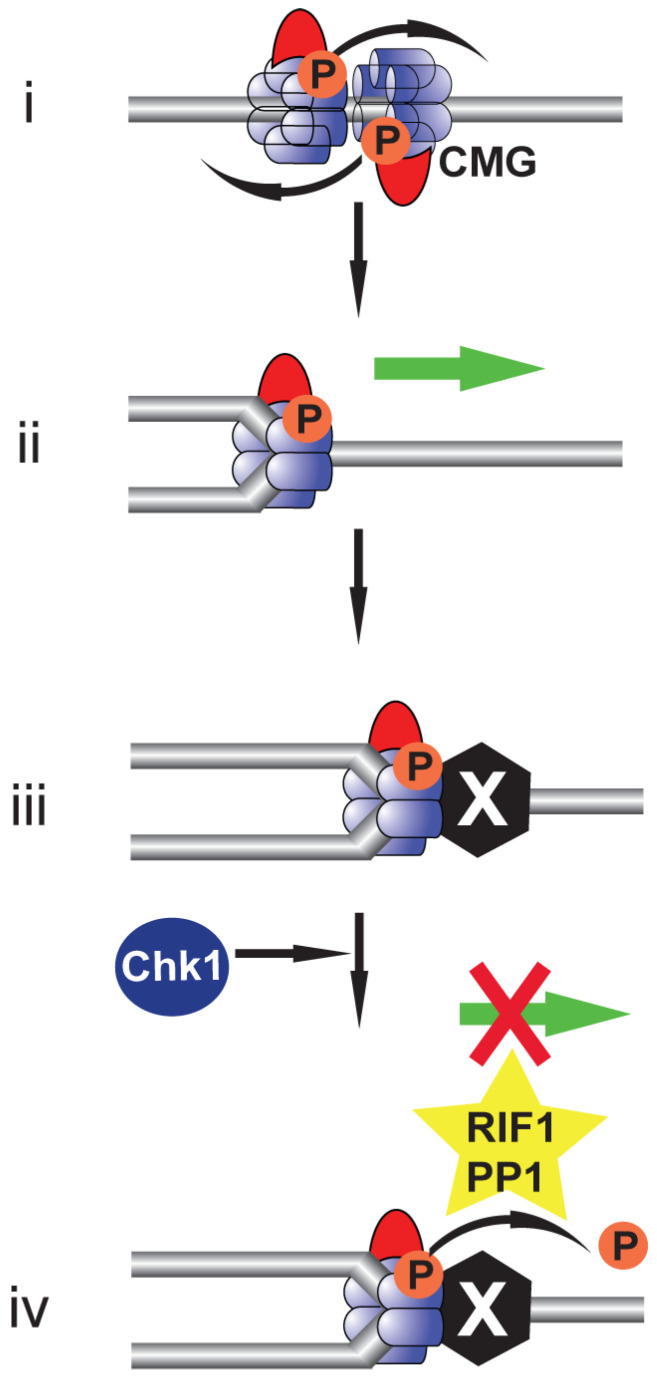
Rif1:PP1-mediated CMG dephosphorylation in response to checkpoint activation inhibits replication fork progression. Cartoon illustrating Rif1:PP1-mediated dephosphorylation of CMG at an elongating replication fork following checkpoint activation. (**i**) Following replication origin activation and CMG formation, the active helicases overtake one another to support bidirectional DNA replication elongation. (**ii**) The maintenance of CMG phosphorylation is required to ensure fork progression. (**iii**) During replication elongation the progress of the replication fork can be stalled upon encountering a site of DNA damage or a protein barrier. (**iv**) In response to replication fork stalling checkpoint activation stabilises the Rif1:PP1 interaction; Rif1:PP1 mediates CMG dephosphorylation promoting the dephosphorylation of specific MCM2 residues but only partial dephosphorylation of MCM4. In the presence of both Rif1:PP1 and Chk1, fork progression is inhibited and the fork remains stable following checkpoint activation; however, in the absence of Chk1, Rif1:PP1 promotes replication fork instability.

**Figure 4 biology-11-00877-f004:**
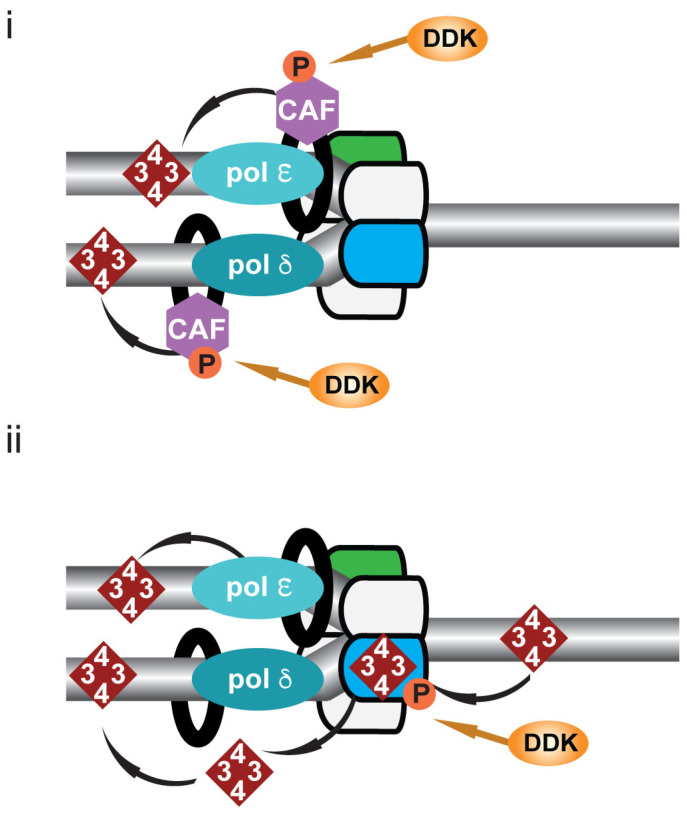
DDK and post-replicative chromatin re-formation. Cartoon illustrating DDK-mediated chromatin re-formation following DNA replication. (**i**) Chromatin assembly factor 1 (CAF-1), a DDK substrate in both budding yeast and higher eukaryotes, binds PCNA (black ring) and deposits H3:H4 tetramers (red diamond) behind the replication fork on newly synthesized DNA. (**ii**) Parental H3:H4 tetramers are distributed to the lagging strand by MCM2 (blue), a DDK substrate and to the leading strand by DNA polymerase ε.

**Table 1 biology-11-00877-t001:** Matched nomenclature for protein names in human and *Xenopus laevis* with those of budding yeast, *Saccharomyces cerevisiae*, fission yeast, *Schizosaccharomyces pombe* and fruit fly, *Drosophila melanogaster*.

Human & *Xenopus laevis*	*S. cerevisiae*, *S. pombe* & *D melanogaster*
**Cdc7**	Cdc7 (Sc), Hsk1 (Sp), Cdc7 (Dm)
**Dbf4, ASK (Hs)**	Dbf4 (Sc), dfp1, him1, rad35 (Sp), Chiffon A (Dm)
**Drf1, Dbf4B (Hs)**	-
**Treslin**	Sld3 (Sc)
**MTBP**	Sld7 (Sc)
**RecQ4**	Sld2 (Sc)
**Claspin**	Mrc1 (Sc)
**Timeless**	Tof1 (Sc)
**Tipin**	Csm3 (Sc)
**Chk1**	Rad53 (Sc)

## Data Availability

Not applicable.

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
