# Peer review of "DDK: The Outsourced Kinase of Chromosome Maintenance"

_biology, 2022, doi:10.3390/biology11060877_

Round 1

Reviewer 1 Report

This is an excellent review that integrates historical and recent findings on the role of DDK in origin activation and MCM phosphorylation, with other activities of DDK in DNA replication related processes such as fork stability, replication stress responses and chromatin reconstitution. It integrates information from both model organisms such yeast and xenopus with information from studies in mammalian cells in a fair, cohesive and clear manner. It describes a model of how origins are selected, involving Treslin:MTBP and integrates this with the established model of origin activation by the coordinate activity of DDK, CDK and the opposing action of phosphatases.

Overall, this work is an important and timely contribution to the field.

Comments and suggestions

1)    A very recent publication by Suski et al  (PMID: 35508654; Nature; 2022; May; 605:357-365; CDC7-independent G1/S transition revealed by targeted protein degradation) suggests that DDK may be dispensable for DNA replication and proliferation in several mammalian cell types, despite the fact that CDC7 is a well-established core essential gene defined by more than 1000 CRISPR/Cas9 screens (https://depmap.org/portal/ ). It also indicates that DDK activity can be bypassed by CDK1 in the xenopus egg extract DNA replication system.

This paper thus challenges the current model of regulation of origin activation by CDK and DDK which is an important part of this review. It would be important to have the authors opinion on how and/or if these recent findings can be reconciled with previous work.

2)    Lines 319-322: revise references. Kohler et al 2019 identified a CDK8/CDK19/CCNC binding domain in MTBP1 while Ferreira et al. (PMID: 33608586, MTBP phosphorylation controls DNA replication origin firing) describes CDK mediated phosphorylation of human MTBP1 in vitro and in immunoprecipitated material from cells. Thus, while it is very likely that CDK8/CDK19/CCNC phosphorylates MTBP1 in intact cells this has not yet directly proven.

3)    Lines 339-352: The section discussing the role of DDK in replication fork stability would benefit in clarity if the authors would better define the concept of replisome instability either at the molecular and/or functional level. i.e fork collapse and DNA damage, loss of replisome proteins from replicating chromatin or incapability to restart from a stalled fork.

4)    Lines 518-520: Additional agents are currently in clinical trials but not mentioned: LY3143921 and TQB3824 (clinicaltrials.org)

Minor & Typos

1.     Figure 1: Blue oval inside green rectangle – this should represent E2F mediated regulation of CDC7  and DBF4 expression. For clarity this blue oval should be labelled “E2F”.

2.     Line 56: Full stop missing at the end of the sentence.

3.     Line 48 and 96: Citations in bold, but not for all other citations in the review.

4.     Line 256: (Figure2) in bold, but not for all other (Figures) in the review.

Author Response

Dear Reviewer,

Thank you very much for your comments.

We have taken account of all of the comments (major and minor) made:

  1. We have added a paragraph accounting for the Suski et al paper at the end of section on Rif1, section 5
  2. We have clarified the text referring to the Boos lab citation
  3. We have added in text to describe fork instability at the beginning of section 9.
  4. The 2 new anti-DDK compounds undergoing clinical trial are now referred to in the text.
  5. All minor comments have been addressed.

Many thanks,

Peter

Reviewer 2 Report

This review titled ‘DDK: The Outsourced Kinase of Chromosome Maintenance’ is entirely dedicated to the roles of the Dbf4-dependent kinase (DDK) in DNA replication, post-replicative chromatin reestablishment, sister chromatid cohesion, replisome stabilization and DNA repair. The review is thorough, detailed, up to date, well written and conveys the idea that cyclin-dependent kinase (CDK) and DDK have intertwined functions, with the former playing a more global role and the latter a more local role in chromosome maintenance. It provides readers with ample literature and useful figures/tables.

The main merits are that authors:

-        emphasize molecular mechanisms and phosphorylation events for which there is demonstrated functional significance

-        integrate and compare data from different organisms, yeasts, xenopus, mammals, and thus focus on conserved mechanisms

-        describe precisely the well-demonstrated mechanism of origin activation by DDK and CDK, while also presenting more speculative functions of DDK after origin firing. The latter part is unfortunately more difficult to read for the non-specialist.

There are however several points where the manuscript could be improved:

1.     Table I with ortholog names should include all proteins shown in figure 2, i.e. also TopBP1/Dpb11, Cdc45 and GINS subunits

2.     Since this table helps readers to navigate through organisms, the authors should refrain themselves of constantly citing orthologs in the text, for better readability (e.g. lines 218, 220, 266, 271, 301…).

3.     Figure 1 is oversimplified and misleading as it suggests that the same CDK species both inactivates Rb, thus enabling DDK expression, and activates origins after priming by DDK. This places CDK both upstream and downstream of DDK, but in fact G1-CDK is upstream and S-CDK is downstream of DDK, functionally, for replication origin firing. This should be corrected, or at least mentioned.

4.     In Figure 2, it is not clear whether step (iii) really exists in cells as it is prevented by CDK. If there is a less-favoured route, it should be denoted by slim or dashed arrows. What triggers RecQ recruitment in step (vi) is not indicated. Why is RecQ phosphorylation by CDK not illustrated by a P sign? Is there any reason not to make step (vàvi) reversible by phosphatase like step (iiiàvi)?

5.     The review would be greatly enriched by the discussion of a recent paper (Suski et al., Nature 605:357) claiming that Cdc7 is not required for DNA replication and cell division in several animal cells, which is at odds with the mainline of this review. This could be done in the context of the discussion of the Cdc7 bypass suppressors in yeast, mcm5-bob1 and MCM4-∆Nter, which make Cdc7 dispensable (line 210). The viability of such bypass mutations also raises the question of the importance of the other portrayed functions of DDK in chromatin reformation, cohesion establishment, replisome stability and DNA repair.

Minor points:

-        Line 48: references in bold

-        Line 102: replace “higher eukaryotes” by “vertebrates”

-        Line 112: to Dbf4, which

-        Line 362: “It has been shown”

-        Line 386: …ATR or ATM, which…

-        Line 478: add “parental” before “histone H3-H4 tetramers”

-        Line 516: …inactivation of p53…

-         

Author Response

Dear Reviewer,

Thank you very much for your comments on our manuscript.

  1. We have added TopBP1 and it's ortholog names to the table. Since the intention of the table is translate between Human/Xenopus nomenclature and that of other organisms we have not added Cdc45 and GINS as the names are conserved between species.
  2. We appreciate the reviewer's point about the repeated mention of 'ortholog' in the text. We have maintained the first mention made for all proteins but removed all subsequent repeats.
  3. This is a very good point. We have updated the figure to take account of this. 
  4. There is evidence that step 3 occurs in the Xenopus cell-free system so until data supporting a preferred pathway is obtained we feel both are viable and should not be distinguished. We have added phosphorylation marks on RecQ4 as suggested; it is the binding of TopBP1 that supports this association. Since Treslin-MTBP phosphosphorylation facilitates complex formation with TopBP1 and step 5-to-6 details the binding of the preformed complex to chromatin bound MCM-P we would predict that should dephosphorylation of T-M occur on chromatin only TopBP1 would come off so the chromatin would return to step 3 not 5. 
  5. As suggested we have added a paragraph about the recent Suski paper. Since we explain this in the context of possible Rif1 inactivation we have added it at the end of section 5 of Rif1.
  6. We have addressed all minor comments

Many thanks for your comments,

Peter

Reviewer 3 Report

Focusing on the role of Dbf4-dependent kinase, this review is also an illuminating perspective on the replication process.

Minor spelling and suggestions

lines 48 and 96: references in bold

line 62: the acronym MTBP is not defined

lines 167-168: N-termini, N termini, instead of N-terminus

line 192: the acronym BRCT is not defined

line 256: (Figure 2) in bold

line 276: Fig instead of Figure

Chapter: Beyond Replication Initiation: the role of DDK in Replication Fork Stability During Replication Stress - A cartoon to visualise the role of Rif1:PP! in restricting fork progression following checkpoint activation would greatly improve the impact of the review

Chapter: DDK & Post-replicative Chromatin Formation - Also here, a cartoon on the effect of DDK-mediated histone chaperon phosphorylation may help to memorize the message

Author Response

Dear Reviewer,

We thank you for comments on our manuscript. We have added two new figures as suggested and corrected the minor errors,

Many thanks,

Peter